# OFFLINE TRACKING WITH OBJECT PERMANENCE

## ABSTRACT

To reduce the expensive labor cost for manual labeling autonomous driving datasets, an alternative is to automatically label the datasets using an offline perception system. However, objects might be temporally occluded. Such occlusion scenarios in the datasets are common yet underexplored in offline auto labeling. In this work, we propose an offline tracking model that focuses on occluded object tracks. It leverages the concept of object permanence which means objects continue to exist even if they are not observed anymore. The model contains three parts: a standard online tracker, a re-identification (Re-ID) module that associates tracklets before and after occlusion, and a track completion module that completes the fragmented tracks. The Re-ID module and the track completion module use the vectorized map as one of the inputs to refine the tracking results with occlusion. The model can effectively recover the occluded object trajectories. It achieves state-of-the-art performance in 3D multi-object tracking by significantly improving the original online tracking result, showing its potential to be applied in offline auto labeling as a useful plugin to improve tracking by recovering occlusions.

## 1 INTRODUCTION

Supervised deep learning-based models have achieved good performance in autonomous driving. However, it usually requires a huge amount of labeled data with high quality to train and tune such data-hungry models. The traditional approach is manual labeling, where all the labels are provided by human annotators. A more effective way is to auto label datasets, where labels can be automatically provided by a trained perception system. Waymo first proposed to auto label data offline to improve the quality of the generated labels Qi et al. (2021). In online tracking, the location of an object is inferred only from past and present sensor data. Offline multi-object tracking (MOT) is acausal and the position of an object can be inferred from past, present, and future sensor data. A consistent estimate of the scene can thus be optimized globally using the data not limited to only a short moment in the past. Based on global information, Yang et al. (2021); Qi et al. (2021); Fan et al. (2023); Ma et al. (2023) have developed offline auto labeling pipelines that generate accurate object trajectories in 3D space from LiDAR point cloud sequence data.

Previous offline labeling methods mainly focused on using the observation sequence to refine the unoccluded trajectories (e.g. location, orientation, and size of the bounding boxes). However, object trajectories and point cloud sequences are sometimes partially missing because of the temporary occlusions on the objects. Severely occluded objects are likely to be missed by detectors and thus cannot be continuously tracked. Furthermore, the trackers in Yang et al. (2021); Qi et al. (2021) adopt the death memory strategy to terminate the unmatched tracks, which means a track not matched with any detections for a predefined number of frames will be terminated. Therefore, the identity of the same object may switch after reappearance from occlusion. Wang et al. (2021) proposed to capture object permanence by not terminating any unmatched tracks. It has been applied in Fan et al. (2023); Ma et al. (2023) for its effectiveness in reducing identity switches (IDS). However, their data association is simply done by using the constant-velocity prediction from a Kalman filter Kalman (1960) which cannot capture the nonlinear motion under long occlusions. Thus, tracking under occlusion remains a challenge in point cloud-based offline auto labeling. Motion prediction models Liang et al. (2020); Gao et al. (2020), on the other hand, can produce accurate vehicle trajectories over longer horizons based on a semantic map. The lanes on the map serve as a strong prior knowledge to guide the motion of target vehicles and thus can be used to estimate motion under occlusion.

In this work, we propose an offline tracking model containing three modules to tackle the afore-mentioned problems. The first module is an off-the-shelf online tracker which produces the initial tracking result. Leveraging the idea of object permanence, the second Re-ID module tries to reassociate the terminated tracklets with the possible future candidate tracklets. Based on the association result, the last track completion module completes the missing trajectories. Taking the insight from the motion prediction task, we have extracted map information as a prior to enhance the association and track completion modules. Unlike common motion prediction methods with a prediction horizon predefined during training Deo et al. (2022); Gilles et al. (2021), our model can use a flexible prediction horizon at inference time to decode the predicted poses in the track completion module, to deal with variable occlusion durations. We train and evaluate our modules on the nuScenes dataset Caesar et al. (2020). Previous offline auto labeling methods are mainly optimized for the precision of the visible bounding boxes, which is a metric for the object detection task. Their performances on MOT metrics are not fully optimized or evaluated. On the other hand, our work focuses on the optimization and evaluation for the MOT metrics (e.g. AMOTA). We observe significant improvements after performing Re-ID over the occlusion scenarios. We then demonstrate the ability of the track completion module to recover the occluded tracks. To further evaluate our model on a larger amount of occlusion cases, we also introduce artificial pseudo-occlusions by masking the ground truth (GT) tracks. We finally evaluated the two modules separately on the pseudo-occlusions.

The contribution of this paper can be summarized as follows:

- We propose an offline tracking model for Re-ID (Sect. 3.1) and occluded track recovery (Sect. 3.2) to track occluded vehicles.

- We innovatively apply prediction-based methods in offline tracking and use the lane map as a prior to improve the Re-ID (Sect. 3.1.2) and track completion (Sect. 3.2). To deal with variable occlusion duration, we decode trajectories from variable time queries (Sect. 3.2).

- We optimize our method for the MOT task and demonstrate the improvements relative to the original online tracking results on the nuScenes dataset (Sect. 5.1), showing the potential of our model to be applied in offline auto labeling as a useful plugin to improve tracking and recover occlusions.

## 2 RELATED WORK

### 2.1 TRACKING UNDER OCCLUSION

Estimating a target's motion under occlusion is one of the main challenges in the tracking task. A common method adopts a constant velocity Weng et al. (2020); Wang et al. (2021) or constant acceleration Wang et al. (2022) transition model and uses the Kalman filter Kalman (1960) to update the tracks when detections are missed due to occlusion. Yin et al. (2021) directly estimates displacement offset (or velocity) between frames. The unmatched track is updated with a constant velocity model. PermaTrack Tokmakov et al. (2021) supervises the occluded motion with pseudo-labels that keep constant velocity. Those methods heavily rely on motion heuristics. Such heuristics work well for short occlusion since objects are likely to keep moving forward with little velocity or acceleration variation. However, they cannot capture the nonlinear motion under long occlusion. Due to the large deviation error accumulated over time, the prediction of the unmatched track would not be associated with the correct detection. Most tracking models that have finite death memory Yin et al. (2021); Weng et al. (2020) would terminate such unmatched tracks, even if the object is detected again later. Such premature termination would lead to IDS. Immortal Tracker Wang et al. (2021) effectively reduces such IDS by extending the life of unmatched tracks forever. Büchner & Valada (2022) uses a multi-modal Graph Neural Network (GNN) to produce robust association offline. Our model does not rely on any handcrafted motion heuristics to predict future motion or to associate. Given any pair of future and history tracklets, the model outputs the learned affinity scores directly.

### 2.2 OFFLINE AUTO LABELING FROM LIDAR POINT CLOUDS

Manually annotating lidar point cloud data takes much effort due to the sparsity of the data and the temporal correlation of the sequence. Several works have attempted to tackle this problem by automatically labeling the dataset. Auto4D and 3DAL Yang et al. (2021); Qi et al. (2021) fully automate annotation by taking initial tracklets generated from an online tracker (i.e. AB3DMOT Weng et al. (2020)). Then, the point cloud features are extracted globally on the temporal horizon to refine the

initial tracks. Given full observation, such refinement utilizes future information to perform global optimization offline. In Auto4d Yang et al. (2021), two rounds of refinement are performed to sequentially refine the box size and trajectory. Point cloud features and motion features are used to jointly refine the trajectory. In 3DAL Qi et al. (2021), static objects and dynamic objects are treated separately. Segmented foreground point cloud feature sequences and bounding box sequence features are used together to refine the box sequences. Though these methods can produce accurate bounding boxes, their data association is still simply done online. The global information available in the offline setting has yet to be utilized in tracking. Such methods are likely to produce IDS and inconsistent tracks under occlusion. CTRL Fan et al. (2023) uses the Immortal Tracker Wang et al. (2021) to associate fragmented tracks due to long occlusion or missing detection. Then it backtraces the tracks by extrapolation to track the missing detections. Detzero Ma et al. (2023) also uses the Immortal Tracker to perform bidirectional tracking on the time horizon and then ensemble the results with forward and reverse order tracking fusion. As a result, CTRL and Detzero have achieved better results which surpass the human annotation. However, their data association still relies on a Kalman filter with the constant-velocity model.

### 2.3 MAP-BASED PREDICTION

Semantic maps are widely used as an input in motion prediction methods. As it contains accurate scene context from the surrounding environment, including road lanes, crosswalks, etc. With such information, models can predict how target vehicles navigate in the environment over a long horizon. HOME Gilles et al. (2021) rasterizes the map as multiple vector layers with distinct RGB values representing different map elements. Alternatively, VectorNet Gao et al. (2020) proposes the vectorized approach which represents curves as vectorized polylines. It uses a subgraph network to encode each polyline as a node in a fully connected global interaction graph. It achieves better performance over the CNN baseline and reduces the model size. To capture higher resolution, LaneGCN Liang et al. (2020) uses polyline segments as map nodes. It models a sparsely connected graph following the map topology. Similarly, PGP Deo et al. (2022) constrains the connected edges such that any traversed path through the graph corresponds to a legal route that a vehicle can take.

## 3 METHOD

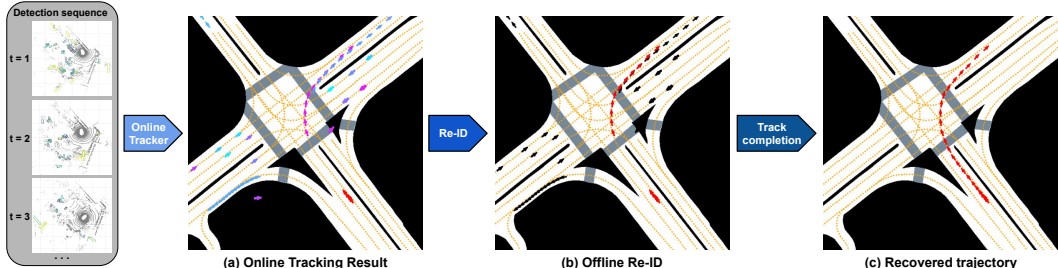

Figure 1: A brief overview of the offline tracking model. **(a) Online tracking result**: Each tracklet is represented by a different color (the history tracklet is red). **(b) Offline Re-ID** : The matched pair of tracklets are red. The unmatched ones are black. **(c) Recovered trajectory** .

As shown in Fig. 1, our model initially takes the detections from a detector as input. Following Wang et al. (2021), we also perform non-maxima suppression (NMS) on the detections. We only perform intra-class NMS with an IoU threshold of 0.1. Then it uses an off-the-shelf online tracker to associate detections and generate initial tracklets. Next, the Re-ID module tries to associate the possible future tracklets with the terminated history tracklet. If a pair of tracklets are matched, the track completion module interpolates the gap between them by predicting the location and orientation of the missing box at each timestamp. Both modules extract motion information and lane map information to produce accurate results. The model finally outputs the track with refined new identities and completes the missing segments within the tracks.

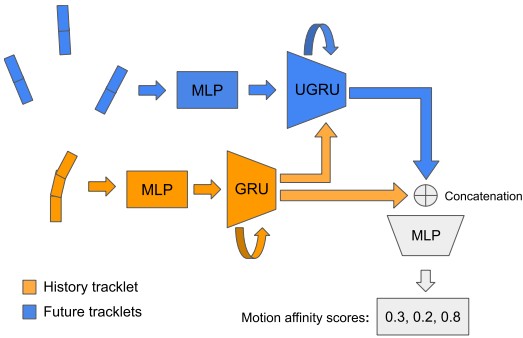

Figure 2: The network structure of motion affinity branch. The orange branch represents the history tracklet encoder, whereas the blue branch represents the future tracklets encoder. Three possible future candidates correspond to the three outputted motion affinity scores.

## 3.1 RE-ID

Based on the concept of object permanence, we assume any tracklets could potentially be terminated prematurely due to occlusion, and any other tracklets that appeared after the termination could be matched. The Re-ID model has two branches, the motion and the map branch. The motion branch takes tracklet motion as input and outputs affinity scores based on vehicle dynamics. The map branch takes motion and the lane map as inputs and outputs affinity scores based on map connectivity. Thus, for each target history tracklet, the model computes motion affinity and map-based affinity scores for it with all the possible future tracklets. The problem can thus be treated as a **binary classification task** for each single matching pair. The final score is a weighted sum of the motion and map affinity scores. We perform bipartite matching by greedily associating history tracklets with their future candidate tracklets based on the final scores. The matching pair which has a tracking score lower than a threshold is excluded from the association. The detailed formulation and input features are introduced in Appendix D.1.

### 3.1.1 MOTION AFFINITY

To compute the motion affinity, we transform the trajectories from the global frame to the local agent frame, where the origin aligns with the pose of the last observation of the history tracklet. The motion feature of each tracklet contains the location, orientation, time and velocity information at every time step. Given one history tracklet and all its possible future matching candidate tracklets, the model takes the motion features as input and outputs affinity scores in the range of $[0, 1]$ for all future tracklet candidates. As shown in Fig. 2, the motion affinity branch has a simple structure that only contains MLP and GRU. The GRU encoder encodes the history motion and outputs the last hidden state as history encoding. The history encoding is then used as the initial hidden state for the UGRU Rozenberg et al. (2021) to encode future motions so that the future encoder is also aware of the history information when encoding the future motion. Unlike the normal U-RNN used in motion prediction for encoding history motion features, our UGRU first does a forward pass and then does a backward pass in reverse order to encode the future motion features. The future motion encodings are concatenated with the history encoding to decode motion affinity scores.

### 3.1.2 MAP-BASED AFFINITY

To improve the association accuracy, we further extract information from the lane graph as another branch. The lane centerlines provide information about the direction of traffic flow and the possible path for the vehicles to follow, which can be utilized for occluded motion estimation over a long horizon. We divide the lanes into sections with a maximum length of 20m. Each lane section is encoded as a node in the graph. For each node, the corresponding lane section is discretized to several poses with a resolution of 1m. Each lane pose feature contains the location, orientation, and semantic information of the lane.

As shown in Fig. 3, the map branch starts by encoding the history and the future tracklets, using the same encoder previously introduced in the motion branch (Sect. 3.1.1). The lane pose features are also encoded in parallel using a single-layer MLP. The model then performs agent-to-lane attention

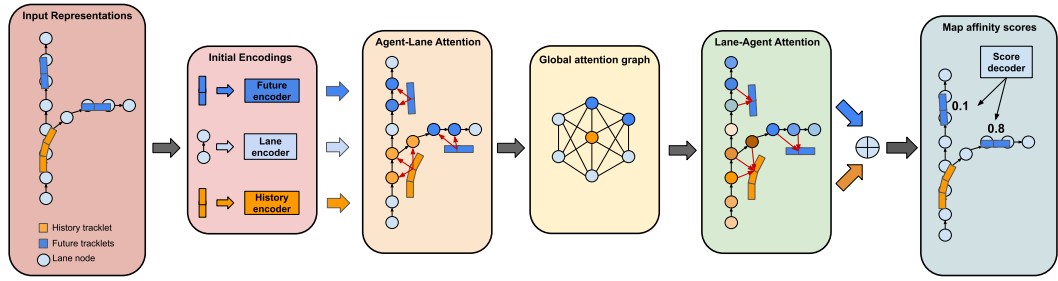

Figure 3: A brief overview of the map branch. The branch starts with three parallel encoders which encode the future tracklet, lanes and history tracklet respectively. The model then propagates information between tracklets and the lane map by performing attention. Finally, map-based affinity scores are decoded from the tracklet features.

to fuse the information from the tracklets to the lanes. The model selectively performs attention such that the tracklet encodings are only aggregated to the surrounding lanes nearby. We adopted the spatial attention layer in Liang et al. (2020) to aggregate the feature from tracklets to lane poses. Then, the model aggregates the feature from each sub-node lane pose to get the lane node features. We apply a 2-layer-GRU to aggregate sub-node features. The first GRU layer is bidirectional so that the information can flow in both directions. After getting the lane node features carrying the information of the tracklets, the model performs gloabl attention Vaswani et al. (2017) to propagate information globally in the graph. Next, the model aggregates the updated lane features back to tracklets by applying a lane-to-agent attention layer Liang et al. (2020). Finally, the future and history tracklet encodings updated with global map information are concatenated together. An MLP decoder is applied to the concatenated features to decode map affinity scores.

### 3.1.3 ASSOCIATION

To match $n$ history tracklets with $N$ future tracklets, we have an affinity score matrices $C_{motion}$ and $C_{map} \in n \times N$ from the two branches, we first filter the matching pairs whose affinity scores are lower than a threshold, then we get the final matching score matrix:

$$C = w \cdot C_{map} + (1-w) \cdot C_{motion} \qquad (1)$$

where $w \in [0,1]$ is a scalar weight. During experiments, we simply set $w$ to 0.5. $C$ is the weighted sum of the two affinity matrices. Finally, we solve the bipartite problem by maximizing the overall matching scores. Due to the sparsity of $C$, we simply perform a greedy matching.

### 3.2 TRACK COMPLETION

With the upstream Re-ID results, we have associated multiple tracklet pairs together. Therefore, we now have multiple fragmented tracks which have a missing segment in the middle. The track completion model interpolates the gaps within the tracks. Based on the history and future motion, the model first generates an initial trajectory in between. Then, a refinement head is applied to refine the initial trajectory based on motion and map features. The trajectory completion model learns a **regression task** as in most motion prediction methods Liang et al. (2020); Deo et al. (2022). The detailed formulation is in Appendix D.2.

We first transform the future and history tracklets from the global frame to the local agent frame. However, instead of using the last pose of the history tracklet as the origin, we set the origin as the midpoint of the line connecting the two endpoints of the missing segment to reduce the over-dependency on the history tracklet. Most prediction models generally have a fixed prediction horizon Liang et al. (2020); Deo et al. (2022). But the occlusion horizon in reality is never fixed. We thus use time features as input queries to decode poses at all the target timesteps. Specifically, each time query feature at time $t$ is given as $[t, t/T]$, where $T$ is the total prediction horizon. So that we can have a variable number of time queries corresponding to a variable prediction horizon.

The overall structure of the track completion model is shown in Fig. 4. It encodes time features as queries and iteratively aggregates context information to the query features to refine the generated

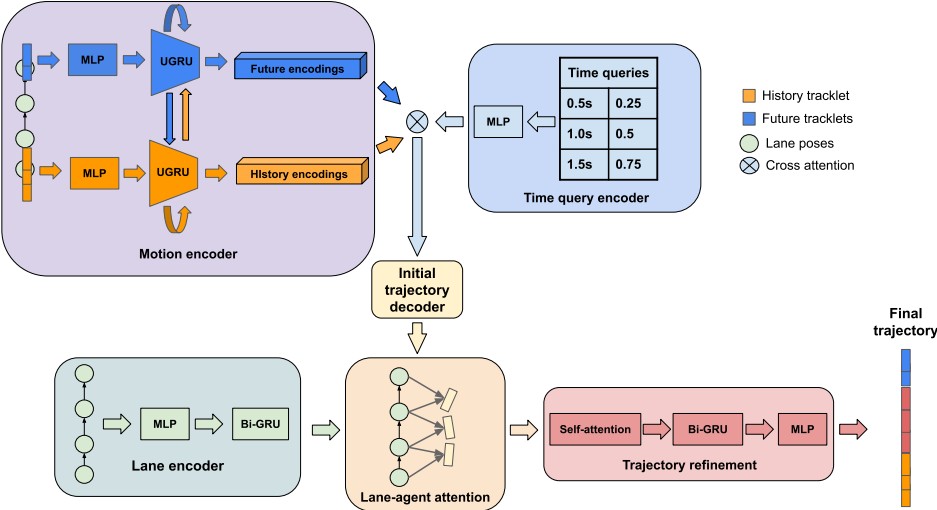

Figure 4: A brief overview of the track completion model. The model stacks several sub-modules together to sequentially aggregate features and refine the results.

tracks. The motion encoder first encodes the history and future motion features using two parallel encoders. The UGRU in each encoder takes the final hidden state from the other, hence the information can propagate through the temporal gap. The time features are encoded with a single-layer MLP as initial queries. A cross-attention layer then aggregates the motion information to the time queries. We also add a skip connection to concatenate the attention output with the motion encoding and time features. To improve the result, we added a refinement phase to the initial trajectory. The model encodes lane features and performs lane-to-agent attention Liang et al. (2020) to fuse the extracted lane information with the generated trajectory information. The trajectory encodings carrying the lane information then go through the final trajectory refinement head which uses a self-attention layer with a skip connection to propagate information within the track. The following bidirectional GRU implicitly smooths the track by aggregating features from each pose to its adjacent poeses. Finally, an MLP decodes the refined trajectory.

### 3.3 LOSS FUNCTION

The Re-ID model and the track completion model are trained separately. Since the losses are commonly used, we show them in Appendix E.

For the Re-ID model, we adopt the focal loss in Lin et al. (2017) to train the binary classification task. In each sample, we only have one GT future tracklet as a positive match, and there are usually multiple negative pairs. We thus use the focal loss for its capability to deal with class imbalance.

For the track completion model, we use the smooth $\ell_1$ loss as in Ye et al. (2022); Liang et al. (2020) to train the coordinates regression head in both the initial trajectory decoder and the final refinement blocks. For yaw angle regression, we first adjust all the GT yaw angles such that their absolute differences with the predicted yaw angles are less than $\pi$. Then, we apply $\ell_1$ loss on the regressed yaw angles. The final regression loss $\mathcal{L}_{\text{reg}}$ is a weighted sum of the coordinates regression loss $\mathcal{L}_{\text{coord}}$ and the yaw angle loss $\mathcal{L}_{\text{yaw}}$. The weights are $\alpha_{coord} = 1.0$ and $\alpha_{yaw} = 0.5$.

### 4 EXPERIMENTAL SETUP

### 4.1 TRAINING SETUP

**Training data:** While Qi et al. (2021); Yang et al. (2021) train their methods using online tracker outputs, we train only using the GT, but additionally inject pseudo-occlusions by masking a partial segment of each GT track. This gives two advantages: **1) More accurate inputs**: Unlike the imperfect trajectories generated from an online tracker, the trajectories from the available autonomous driving datasets are accurate and contain little noise. With proper data augmentation, our model can

still deal with imperfect data when inferencing with the trajectories produced from an online tracker. **2) More training samples**: Since we have longer intact tracks as input, we can generate data with longer occlusions.

**Pseudo-occlusion:** We randomize the length of the pseudo-occlusion for each track during the training of the Re-ID and track completion models. The pseudo-occlusion duration is selected such that the history tracklet and all the future tracklets have at least one observation.

**Data augmentation:** To deal with the possible imperfections from the online tracking result during inference, we augment the data during training by randomly rotating the local frame in each sample and adding random noise to the motion inputs.

## 4.2 EVALUATION SETUP

We train and evaluate our model on the nuScenes dataset Caesar et al. (2020). The standard MOT evaluation on nuScenes could punish the fully occluded predictions since the evaluation code filters out all the GT boxes with no points inside and then linearly interpolates the GT tracks. Therefore, some of the bounding boxes recovered from occlusion could be regarded as false positives (FP) as the GT boxes they are supposed to be matched with are linearly interpolated. We thus train and validate our model on the training split of the nuScenes dataset. The evaluation is done on the validation split instead of the test split so that we can adjust the evaluation procedure for occlusion cases. In each of the experiments below, we use one of the following three evaluation setups as indicated.

**Official nuScenes setup**: Following the official nuScenes MOT evaluation protocol, we filter out all the empty GT boxes without any lidar or radar points inside. We thus focus on the visible GT boxes. The evaluation result is thus comparable to the results shown on the nuScenes leaderboard.

**All-boxes setup**: To get a comprehensive evaluation result, we do not filter any GT boxes. Therefore, the evaluation takes both the visible and the occluded boxes into account.

**Pseudo-occlusion setup**: To generate more occlusions with longer duration for evaluation, we take the GT vehicle tracks from the nuScenes prediction validation set and mask a partial segment within each track. Unlike the previous setups where the model takes the online tracking result as input, the model takes the unoccluded GT tracklets as inputs under this setup.

## 5 EXPERIMENTS

We first evaluate our offline tracking model on the imperfect data generated from the online tracking result. We show the relative improvements it brings to the original online tracking result on MOT metrics. In this setting, the experiments evaluate the potential of the offline tracking model to be applied in offline auto labeling. We use CenterPoint Yin et al. (2021) as the initial off-the-shelf detector and tracker. Next, we evaluate our offline tracking model using human-annotated data. We mask the vehicle tracks to create a large amount of pseudo-occlusion cases for evaluation. The human-annotated data have little input noise, which excludes the imperfections from the online tracker so the evaluation focuses solely on the offline tracking model we proposed. Furthermore, the real occlusion cases only take a relatively small portion of the dataset compared to the unoccluded cases and the occlusion duration is typically short. Unlike the real occlusions in the dataset, the generated pseudo-occlusions are abundant and typically have longer durations. We focus our method only on the vehicle classes. The implementation details are in Appendix F.

### 5.1 QUANTITATIVE EVALUATION WITH ONLINE TRACKING RESULTS

#### 5.1.1 RE-ID MODEL EVALUATION

In the sole evaluation of the Re-ID model, we adopt the **official nuScenes setup** which follows the standard evaluation protocol of nuScenes and filters out the empty GT boxes for evaluation. We only use the Re-ID model to reassociate the tracklets before and after occlusions, and we do not use the track completion model to interpolate the trajectories between the gaps. Instead, the nuScenes evaluation code automatically does a simple linear interpolation to fill all the gaps on both the GT tracks and the predicted tracks. For comparison, we select the lidar-based SOTA methods that also use CenterPoint Yin et al. (2021) detections from the nuScenes leaderboard.

| Method | Overall | Car | Bus | Truck | Trailer |
|---|---|---|---|---|---|
| CenterPoint Yin et al. (2021) | 69.8 | 82.9 | 71.1 | 59.9 | 65.1 |
| SimpleTrack Pang et al. (2022) | 70.0 | 82.3 | 71.5 | 58.7 | 67.3 |
| UVTR Li et al. (2022) | 70.1 | 83.3 | 67.2 | 58.4 | **71.6** |
| Immortal Tracker Wang et al. (2021) | 70.5 | 83.3 | 71.6 | 59.6 | 67.5 |
| NEBP Satorras & Welling (2021) | 70.8 | 83.5 | 70.8 | 59.8 | 69 |
| 3DMOTFormer++ Ding et al. (2023) | 72.3 | 82.1 | 74.9 | 62.6 | 69.6 |
| ShaSTA Sadjadpour et al. (2023) | 73.1 | 83.8 | 73.3 | **65** | 70.4 |
| Offline Re-ID (Motion + Map) | **73.4** | **84.2** | **75.1** | 64.1 | 70.3 |

Table 1: Comparison of AMOTA scores over the SOTA methods using CenterPoint Yin et al. (2021) detections on the nuScenes test split (official nuScenes setup).

| Method | AMOTP↓ / m | Recall↑ | MOTA↑ | IDS↓ |
|---|---|---|---|---|
| CenterPoint Yin et al. (2021) | 0.596 | 73.5 | 59.4 | 340 |
| SimpleTrack Pang et al. (2022) | 0.582 | 73.7 | 58.6 | 259 |
| UVTR Li et al. (2022) | 0.636 | **74.6** | 59.3 | 381 |
| Immortal Tracker Wang et al. (2021) | 0.609 | 74.5 | 59.9 | 155 |
| NEBP Satorras & Welling (2021) | 0.598 | 74.1 | **61.9** | **93** |
| 3DMOTFormer++ Ding et al. (2023) | 0.542 | 73.0 | 58.6 | 210 |
| ShaSTA Sadjadpour et al. (2023) | 0.559 | 74.3 | 61.2 | 185 |
| Offline Re-ID (Motion + Map) | **0.532** | 74.2 | 61.3 | 204 |

Table 2: Comparison of MOT metrics over the SOTA methods using CenterPoint Yin et al. (2021) detections on the nuScenes test split (official nuScenes setup).

From the results in Tab. 1 and 2, our model effectively improves the original online tracking result from CenterPoint Yin et al. (2021). Please also note that the Re-ID model only refines the limited occlusion cases. Therefore, the improvement brought by the Re-ID model is upper bound by the number of occlusions. Yet the result after refinement already outperforms the other SOTA methods on AMOTA and AMOTP, indicating the Re-ID model can accurately reassociate the tracklets before and after occlusions. We thus have demonstrated the potential of our Re-ID model to improve the offline tracking result during auto labeling, especially for occlusion-rich scenes. We show the effects of the map and the motion branches separately and the general improvements brought by our Re-ID model to other SOTA online trackers on the validation split in Appendix B.

### 5.1.2 TRACK COMPLETION MODEL EVALUATION

Based on the Re-ID results, the track completion model recovers the missing trajectories between all the matched tracklet pairs. Since we want to evaluate the ability of our model to recover the occluded trajectories, we adopt the **all-boxes setup** and thus do not filter out any GT bounding boxes. A detailed description of the experimental setup is in Appendix G.1. The results are shown in Tab. 3. We have shown the improvements our model brings to multiple SOTA online trackers. We have only tuned our model using the results from CenterPoint tracker Yin et al. (2021) and then applied our model to the results from other trackers. The improvements have demonstrated that our model can be used as a general plugin to recover the imperfections caused by occlusions during offline autolabeling. Our model is compatible with any online tracker, thus it can be easily integrated into the existing offline autolabeling frameworks Qi et al. (2021); Yang et al. (2021). We have also shown the ability of our track completion model to recover the occluded objects in Appendix C.

| Track Completion | AMOTA↑ | | AMOTP↓ / m | | IDS↓ | | Recall↑ | |
|---|---|---|---|---|---|---|---|---|
| | w/o | w | w/o | w | w/o | w | w/o | w |
| CenterPoint Yin et al. (2021) | 70.2 | **72.4** | 0.634 | **0.615** | 254 | **183** | 73.7 | **74.5** |
| SimpleTrack Pang et al. (2022) | 70.0 | **71.0** | 0.668 | **0.629** | 210 | **170** | 72.5 | **72.9** |
| VoxelNet Chen et al. (2023) | 69.6 | **70.6** | 0.710 | **0.665** | 308 | **230** | 72.8 | **72.9** |
| ShaSTA Sadjadpour et al. (2023) | 72.0 | **72.6** | 0.612 | **0.593** | 203 | **174** | 73.0 | **75.3** |

Table 3: Track completion evaluation on the nuScenes validation split (All-boxes setup). w: with track completion. w/o: without track completion.

| Method | CVM | Motion | Map | Motion + Map |
|---|---|---|---|---|
| **Association accuracy** ↑ | 58.7% | 90.1% | 89.6% | **90.3%** |

Table 4: Re-ID evaluation on the nuScenes validation split (Pseudo-occlusion setup).

| Method | ADE↓ / m | Yaw error↓ / deg | MR↓ |
|---|---|---|---|
| HOME Gilles et al. (2021) | 0.814 | - | 24.3% |
| Motion | 0.705 | 2.38 | 14.9% |
| Motion + map | **0.667** | **2.23** | **13.3%** |

Table 5: Track completion evaluation on the nuScenes validation split (Pseudo-occlusion setup). The prediction horizon is 6s which is the same as the nuScenes prediction challenge.

## 5.2 QUANTITATIVE EVALUATION WITH PSEUDO-OCCLUSIONS

### 5.2.1 RE-ID MODEL EVALUATION

We evaluate the Re-ID model on pseudo-occlusions following the setup in Appendix G.2. We also include a constant velocity model (CVM) associator as our baseline. The CVM associator takes the last observable position and velocity as inputs then predicts future trajectories with a constant velocity. The future tracklet which has the shortest distance to the constant velocity prediction will be matched. The association accuracies are listed in Tab. 4. Given inputs without noise, our model in all settings achieves high association accuracies over 89% and outperforms the CVM associator by a large margin over 30%. With the perfect motion input, the motion branch has a higher association accuracy over the map branch by $0.5\%$. After the combination of the motion and the map branch, the accuracy increases by $0.2\%$ compared to the result using only the motion branch. Hence, the two branches are complementary to each other.

### 5.2.2 TRACK COMPLETION MODEL EVALUATION

To standardize the evaluation, for each evaluated sample track, we mask 6 seconds of its trajectory and take 2 seconds of history tracklet and future tracklet as inputs. We chose HOME Gilles et al. (2021) as a baseline and re-implemented it on nuScenes. HOME originally decodes the predicted trajectories from sampled endpoints. To make the comparison fair, we directly give the GT trajectory endpoints to the HOME, so that the model is also aware of the future.

From the result, the performance is improved on every metric after using the map information for trajectory refinement, indicating the extracted map prior improves the trajectory recovery. Our model also outperforms HOME on both ADE and MR, demonstrating its potential to recover trajectories under long occlusions.

## 5.3 QUALITATIVE RESULTS

We have visualized several representative samples in the evaluation with the online tracking result and pseudo-occlusions. Due to the page limit, we show them in Appendix A.

## 6 CONCLUSION

While the previous point cloud-based offline auto labeling methods focused on generating accurate bounding boxes over visible objects, we have proposed a novel offline tracking method focusing on the occlusions on vehicles. The Re-ID module can effectively reduce IDS caused by premature termination under occlusion. Based on the Re-ID result, the track completion model recovers the occluded trajectories. Leveraging the idea from motion prediction, we innovatively extracted information from the lane map and used it as a prior to improve the performance of our models. We have demonstrated the ability of our model on the nuScenes validation split, using both the imperfect online tracking results and the handcrafted pseudo-occlusion data. The offline tracking model improves the original online tracking result and achieves SOTA performance, showing its potential to be applied in auto labeling autonomous driving datasets. It also achieves good performances on the handcrafted pseudo-occlusions and outperforms the baselines by large margins. Future work will aim to unify the Re-ID and track completion modules as an end-to-end model.

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

APPENDIX

In this appendix, we first present the qualitative results A. We also show supplementary experimental results of the Re-ID model B and track completion model C on the validation split. Next, give the detailed formulation of the Re-ID and the track completion D. We also give the formulas and the hyper-parameters of the training loss functions E. Then, we describe the implementation details F. Finally, we give a detailed explanation of the experimental setups G.

## A  QUALITATIVE RESULTS

### A.1  JOINT EVALUATION WITH ONLINE TRACKING RESULTS

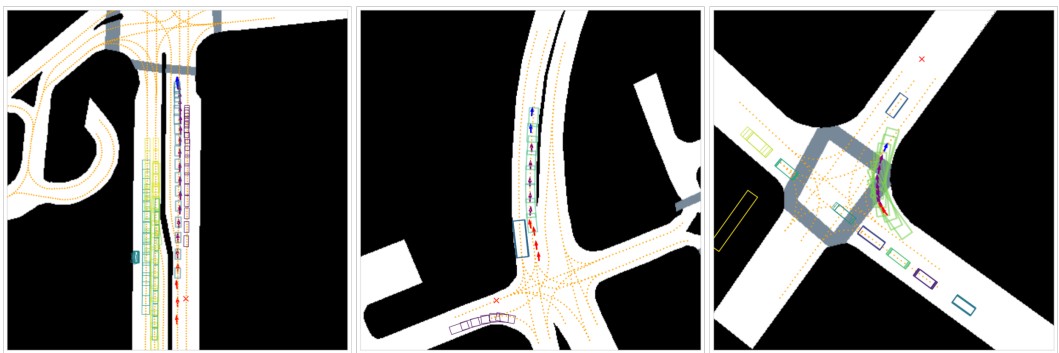

Figure 5: Qualitative results of the offline tracking model. In each sample, GT boxes are plotted as rectangles. Each GT track is represented by a unique color. To separate the output of the model from the GT tracks, we use arrows to represent the output poses from the model instead of rectangles. Red and blue arrows are the history and future tracklets matched by the offline Re-ID model. The recovered trajectories are plotted as purple arrows. The orange dotted lines are the lanes. In the background, the white area is drivable. The gray area in the right plot is the pedestrian crossing. The red cross is the average position of the ego vehicle during the occlusion.

As shown in Fig. 5, we applied our offline tracking model to improve the online tracking result by jointly performing Re-ID and track completion. Three samples in the three plots show different occlusion types. The left plot shows a long occlusion with a duration of 8s and a gap distance of 71m. The middle plot shows a relatively short occlusion with a duration of 3s and a gap distance of 25m. The right plot shows an occlusion that happened on a turning vehicle at a crossroad. The occlusion duration in the third sample is 3.5s and the gap distance is 13m. With the three samples, we show the ability of our offline tracking model to deal with long and short occlusions, as well as the non-linear motion under occlusion. Taking the online tracking results and the semantic map as inputs, our offline tracking model first correctly associated together the tracklets belonging to the same GT track. The associated future (blue) and history (red) tracklets in each plot are covered by a single GT track. Then the model recovered the occluded trajectories by interpolating the gaps between the associated tracklet pairs. The recovered trajectories (purple) also aligned well with the corresponding GT tracks.

### A.2  RE-ID MODEL EVALUATION ON PSEUDO-OCCLUSION

As shown in Fig. 6, a representative case is selected to show the map information extracted from the lane graph can help correct the association. In the left sub-figure, the motion branch tends to assign a high affinity score (red) to the future tracklet candidate above. Though this association fits the motion pattern, it is a false match. After adding the map information, the affinity score of the false future candidate is suppressed (colors become bluer), since it is not on the same lane as the history tracklet. The affinity score of the true future tracklet increases in the map branch (colors become redder), showing that our offline tracking model can utilize the map prior to accurately associating tracklets.

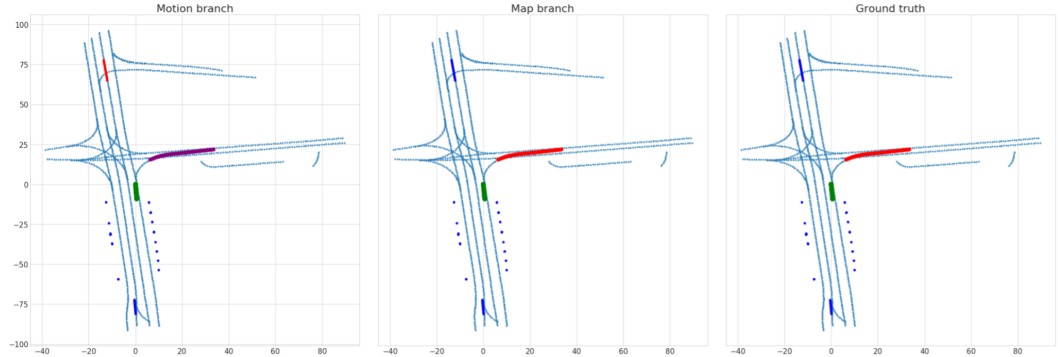

Figure 6: Qualitative results of the Re-ID model evaluated with pseudo-occlusion. The history tracklets are green. Future tracklets are colored according to their affinity scores. Higher scores are represented in red and lower scores are in blue. The thin dotted lines represent the lanes from the semantic map. The left figure shows the prediction from the motion branch. The middle figure shows the prediction from the map branch, and the right figure shows the GT.

### A.3 TRACK COMPLETION MODEL EVALUATION ON PSEUDO-OCCLUSION

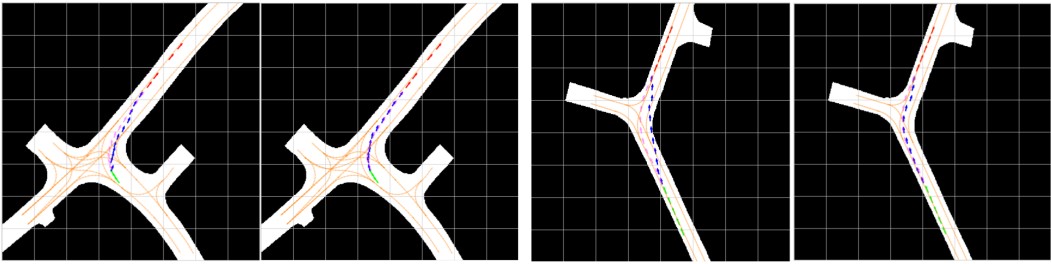

Figure 7: Qualitative results of the track completion model evaluated with pseudo-occlusion. In the plot, 2 pairs of prediction results are shown. In each pair, the left image shows the decoded trajectory using only the motion feature, whereas the right image shows the trajectory after refinement. The history tracklets, future tracklets, predicted trajectories and GT trajectories correspond to the green, red, blue, and pink arrows respectively. The lanes are in orange.

As shown in Fig. 7, both the trajectories before and after refinement can predict the yaw angle very well. However, the initial trajectories before refinement are not smooth sometimes. They constantly deviate from the lane and GT trajectory. After refinement, the trajectories are smoother and tend to follow the lane. Less deviation from the GT trajectory is observed. Hence it accords with our previous quantitative evaluation, showing our track completion model is able to leverage the map prior to generating accurate trajectories.

## B QUANTITATIVE RE-ID RESULTS ON VALIDATION SPLIT

To ablate the effect of motion and map branches, we also separately evaluated our model on the nuScenes validation split. The results are shown in Tab. 6 and 7. The performance of our model generally improves after aggregating with map information. Our model outperforms the baselines on AMOTA, including the offline method 3D-PM-CL-CP which uses camera and lidar data. Please also note that we did not tune our model on the validation split. Also, the refinement is performed only in limited scenarios where occlusion happens. Though the margin between our method and the Immortal Tracker is not significant, we argue this is because the overall AMOTA is a simple macro average over all classes. If the imbalanced box numbers in different classes are taken into account, our model will be more advantageous, since our model can predict the abundant car class (58317 boxes) well, whereas the Immortal Tracker can predict the bus class (2112 boxes) well. Compared

with the input result from the CenterPoint tracker Yin et al. (2021), our model reduces IDS by 45% and increases AMOTA by 2% on the vehicle tracks after refinement.

| Method | Overall | Car | Bus | Truck | Trailer |
|---|---|---|---|---|---|
| CenterPoint Yin et al. (2021) | 70.9 | 84.3 | 83.0 | 69.1 | 47.1 |
| Immortal Tracker Wang et al. (2021) | 72.8 | 84.0 | **87.1** | 69.4 | 50.4 |
| 3D-PM-CP Büchner & Valada (2022) | 71.8 | 84.9 | 83.7 | 68.9 | 49.7 |
| 3D-PM-CL-CP Büchner & Valada (2022)* | 72.4 | 85.1 | 85.5 | 69.5 | 49.3 |
| Offline Re-ID (Motion only) | 72.7 | 84.9 | 85.4 | 69.7 | **50.9** |
| Offline Re-ID (Map only) | **72.9** | 85.2 | 85.5 | **70.0** | 50.8 |
| Offline Re-ID (Motion + Map) | **72.9** | **85.3** | 85.5 | **70.0** | 50.8 |

Table 6: Comparison of AMOTA scores on the nuScenes validation split (official nuScenes setup). * denotes the use of sensor data for tracking.

| Method | AMOTP↓ / m | Recall↑ | MOTA↑ | IDS↓ |
|---|---|---|---|---|
| CenterPoint Yin et al. (2021) | 0.623 | 73.0 | 60.5 | 267 |
| Immortal Tracker Wang et al. (2021) | **0.574** | 73.9 | 60.5 | **109** |
| Offline Re-ID (Motion only) | 0.613 | **74.8** | 62.1 | 168 |
| Offline Re-ID (Map only) | 0.603 | 74.6 | **62.2** | 145 |
| Offline Re-ID (Motion + Map) | 0.603 | 74.6 | 62.1 | 147 |

Table 7: Comparison of AMOTA scores on the nuScenes validation split (official nuScenes setup). * denotes the use of sensor data for tracking.

| Offline Re-ID | AMOTA↑ | | AMOTP↓ / m | | IDS↓ | | Recall↑ | |
|---|---|---|---|---|---|---|---|---|
| | w/o | w | w/o | w | w/o | w | w/o | w |
| CenterPoint Yin et al. (2021) | 70.9 | **72.9** | 0.623 | **0.603** | 267 | **183** | 73.0 | **74.6** |
| SimpleTrack Pang et al. (2022) | 70.6 | **71.4** | 0.637 | **0.619** | 175 | 179 | 71.9 | **73.0** |
| VoxelNet Chen et al. (2023) | 70.3 | **71.1** | 0.690 | **0.653** | 337 | **245** | **74.7** | 73.5 |
| ShaSTA Sadjadpour et al. (2023) | 72.7 | **73.3** | 0.600 | **0.583** | 210 | **180** | **74.2** | 72.5 |

Table 8: Improvements brought by our offline Re-ID model over different SOTA online trackers (official nuScenes setup). w: with Re-ID. w/o: without Re-ID.

In Tab. 8, we also show the improvements our Re-ID model can bring to online trackers. We have only tuned our model using the results from CenterPoint tracker Yin et al. (2021). Still, our model can generally improve the tracking results from other online trackers, which demonstrates the potential of our model to be used as a useful plugin to improve the tracking result in offline auto labeling.

## C  Occluded objects recovery on validation split

In Tab. 9, we show that the track completion model can effectively recover the occluded trajectories by increasing the number of TP boxes and reducing the number of FN boxes. The number of FP is less than using the Immortal Tracker adopted in Ma et al. (2023); Fan et al. (2023).

## D  Detailed formulation

### D.1  Re-ID formulation

The upstream online tracker provides a set of tracklets $\mathbf{T}$ in the scene. In $\mathbf{T}$, history tracklets are the tracklets that end earlier before the last frame of the scene. They are represented as $\mathbf{H} = [H_1, \ldots, H_n]$. For the $i$-th history tracklet $H_i$, it contains history observations of $\left[h^i_{-T_i}, \ldots, h^i_0\right]$, where $h^i_0$ is the last observation of $H_i$ on the temporal horizon and $h^i_{-T_i}$ the first observation. Each observation contains the location, orientation, size, uncertainty, and velocity. It is defined as:

$$h^i_t = \left(x^i_t, y^i_t, z^i_t, \theta^i_t, l^i_t, w^i_t, h^i_t, s^i_t, v^i_{xt}, v^i_{yt}, v^i_{zt}\right) \tag{2}$$

| Method | AMOTA↑ | TP↑ | FP↓ | FN↓ |
|---|---|---|---|---|
| CenterPoint Yin et al. (2021) | 70.2 | 59332 | **8197** | 14704 |
| Immortal Tracker Wang et al. (2021) | 72.3 | 59271 | 9593 | 14883 |
| Offline Track Completion | **72.4** | **60675** | 8953 | **13432** |

Table 9: Our track completion model can effectively recover the occluded objects by increasing TP (All-boxes setup).

where $l, w, h$ represent the size of the $j$-th bounding box. $s$ is the confidence score, and $v_x, v_y, v_z$ the velocities on $x, y, z$ directions.

For each history tracklet $H_i$, it has a set of possible future candidate tracklets for matching: $\mathbf{F}^i = \left[F_1^i, \ldots, F_n^i\right]$. Each future tracklet $F_j^i$ starts after the termination of $H_i$:

$$F_j^i = \left[f_{t_j}^i, \ldots, f_{t_j+T_j}^i\right], \text{ s.t. } t_j > \tau \tag{3}$$

In practice, we set $t_j > \tau$ where $\tau$ is the horizon of the death memory of the online tracker since the GT future tracklets reappearing within such period are likely to be associated by the online tracker. Each tracklet has a feature dimension of $T \times 8$, where $T$ is the length of the tracklet. The 8 features correspond to $[x, y, \theta, t, cos(\theta), sin(\theta), v_x, v_y]$, where $x, y$ are the local BEV coordinates, $\theta$ the yaw angle, $t$ the time relative to the last observation of the history tracklet and $v_x, v_y$ the velocities on x and y direction.

The model also extracts information from the fully connected lane graph $\mathcal{G} = (V, E)$ where each node in $V$ is a lane centerline section. $E$ is the set containing all the edges for each pair of nodes. Each lane node is encoded from a lanelet, which is a section of the lane with a maximal length of 20m. A lanelet is represented as several lane poses. Each lane pose contains the feature of $[x_{lane}, y_{lane}, \theta_{lane}, cos(\theta_{lane}), sin(\theta_{lane}), \mathcal{D}, \mathcal{L}_{lane}]$, where $x_{lane}, y_{lane}$ are the BEV coordinates of the lane poses in the local frame, $\theta_{lane}$ the yaw angle, $\mathcal{D}$ the binary flag indicating whether the lane section ends. Following PGP Deo et al. (2022), we have also included $\mathcal{L}_{lane}$, a 2-D binary vector indicating whether the pose lies on a stop line or crosswalk, to capture both the geometry as well as traffic control elements along the lane. The graph thus has a feature with the dimension of $N_{lane} \times l \times 8$, where $N_{lane}$ is the number of lane nodes and $l$ is the length of a lanelet.

Given a history tracklet $H_i$ and a future tracklet $F_j^i$ and lane graph $\mathcal{G}$, the model outputs motion affinity scores and map-based affinity scores:

$$C_{motion}^{i,j} = Net_{motion}(H_i, F_j^i) \in [0, 1]$$
$$C_{map}^{i,j} = Net_{map}(H_i, F_j^i, \mathcal{G}) \in [0, 1] \tag{4}$$

where $Net_{motion}$ and $Net_{map}$ represent the networks of the motion branch and map branch respectively. The final matching score for the association $C^{i,j}$ is a weighted sum of $C_{motion}^{i,j}$ and $C_{map}^{i,j}$. Therefore, we can construct a matching score matrix $\mathbf{C} \in n \times N$ for tracklet association, where $N$ is the number of tracklets $\mathbf{T}$ in the scene and $n$ the number of history tracklets $\mathbf{H}$.

$$\mathbf{C} = \begin{bmatrix} C^{1,1} & \cdots & C^{1,N} \\ \vdots & \ddots & \vdots \\ C^{n,1} & \cdots & C^{n,N} \end{bmatrix} \tag{5}$$

Each row in $\mathbf{C}$ represents a history tracklet and each column represents a future tracklet. Based on the established affinity matrix, bipartite matching is performed for association such that the sum of the matching scores is maximized.

$$\max_{X \in \{0,1\}^{n \times N}} \sum_{i=1}^{n} \sum_{j=1}^{N} \mathbf{C}_{i,j} X_{i,j} \tag{6a}$$

$$\text{s.t.} \sum_{i=1}^{n} X_{i,j} \leq 1, \forall j \tag{6b}$$

$$\sum_{j=1}^{N} X_{i,j} \leq 1, \forall i \tag{6c}$$

For tracklet pairs violating the constraint Eq. 3, the corresponding elements will not be considered for matching.

## D.2 TRACK COMPLETION FORMULATION

Following the formulation in Sect. D.1, the model receives a history observation sequence $H_i = \left[ h^i_{-T_i}, \ldots, h^i_0 \right]$, and a matched future observation sequence $F_i = \left[ f^i_{t_i}, \ldots, f^i_{t_i+T'_i} \right]$, where $t_i$ is the occlusion horizon of the $i$-th trajectory, $T_i$ and $T'_i$ the lengths of history and future sequence respectively. The model predicts a motion sequence $\mathcal{P}_i = \left[ p^i_1, \ldots, p^i_{t_i-1} \right]$ in between. Here, we choose to output the $x, y$ coordinates in the BEV plane, and the yaw angle $\theta$.

$$p^i_t = Net_{completion}(F_i, H_i, \mathcal{G}, t) = \left[ x^i_t, y^i_t, \theta^i_t \right] \tag{7}$$

where $Net_{completion}$ represents the network of the track completion model. We use the same motion features as in the previous Sect. D.1, with the exception that we do not use the velocity explicitly. This results in a feature dimension of $T_{input} \times 6$. Given that the start and end pose of the predicted are already known, the model can implicitly predict the velocity in between. Hence, we discard the velocity features to reduce the possible noise from the imperfect online tracking result. For the semantic map, we also parameterize the lanes in the same way as introduced in the previous Sect. D.1.

## E LOSS FUNCTIONS

We adopt the focal loss Lin et al. (2017) to train the Re-ID model.

$$\mathcal{L}_{cls}(k_t) = -\alpha_t (1 - k_t)^\gamma \log(k_t)$$
$$k_t = \begin{cases} k & \text{if } y = 1 \\ 1 - k & \text{otherwise} \end{cases} \tag{8}$$

where $k$ is the predicted affinity score, and $y$ is the GT score. $\alpha_t$ and $\gamma$ are two hyperparameters, which are set to 0.5 and 2.0 respectively.

For the track completion model, we use Huber loss (i.e. smooth L1 loss) as in Ye et al. (2022); Liang et al. (2020); Gu et al. (2021) to train the $x, y$ regression heads in both the initial trajectory decoder and the final refinement blocks:

$$\mathcal{L}_{coord} = \frac{1}{t_f} \sum_{t=1}^{t_f} d\left(m_t - m_t^{gt}\right)$$
$$d(x_i) = \begin{cases} 0.5 \|x_i\|_2 & \text{if } \|x_i\| < 1 \\ \|x_i\| - 0.5 & \text{otherwise} \end{cases} \tag{9}$$

where $m_t = [x_t, y_t]$ are the predicted $x, y$ coordinates in local agent frame, $t_f$ the prediction horizon. $\|\cdot\|$ and $\|\cdot\|_2$ denotes the $\ell_1$ and $\ell_2$ norm . For yaw angle regression, we first adjust all the GT yaw angles such that their absolute differences with the predicted yaw angles are less than $\pi$. Then, we apply $\ell_1$ loss on the regressed yaw angles:

$$\mathcal{L}_{yaw} = \frac{1}{t_f} \sum_{t=1}^{t_f} \left| \theta_t - \theta_t^{gt} \right| \tag{10}$$

The final regression loss $\mathcal{L}_{reg}$ is:

$$\mathcal{L}_{reg} = \alpha_{coord} \mathcal{L}_{coord} + \alpha_{yaw} \mathcal{L}_{yaw} \tag{11}$$

where $\alpha_{coord}$ and $\alpha_{yaw}$ are the weights. We set $\alpha_{coord} = 1.0$ and $\alpha_{yaw} = 0.5$.

## F  IMPLEMENTATION DETAILS

We train the Re-ID model for 50 epochs on Tesla V-100 GPU using a batch size of 64 with the AdamW Loshchilov & Hutter (2017) optimizer with an initial learning rate of $1 \times 10^{-3}$, which decays by a factor of 0.6 every 10 epochs. The training time is 11.5 hours. We train the track completion model following the same setting except for a different decay factor of 0.5 every 10 epochs. The training time is 4.2 hours.

Our model is agnostic to the selections of the detector and online tracker. For the evaluation with online tracking results, we use an association threshold of 0.9. If the the motion affinity and map-based affinity scores are both lower than the threshold, the corresponding matching pair would not participate in the association. For the evaluation with pseudo-occlusions, we use the tracks from the nuScenes validation split provided by the nuScenes software devkit for the motion prediction challenge.

## G  SUPPLEMENT OF EXPERIMENTAL SETUPS

### G.1  EXPERIMENTAL SETUP FOR TRACK COMPLETION MODEL EVALUATION WITH ONLINE TRACKING RESULTS

We take the Re-ID result refined by the motion and the map branch as input. Based on the Re-ID result, the track completion model interpolates the gaps and recovers the missing trajectories. If a gap within a track is spatially larger than 3m or temporally longer than 1.8 seconds, it is considered a possible occlusion, and the track completion model predicts the poses for the missing timestamps in between. Otherwise, the evaluation code automatically does a linear interpolation to fill the small gaps. The sizes of the recovered bounding boxes are linearly interpolated from the two bounding boxes in the history tracklet and the future tracklet.

### G.2  EXPERIMENTAL SETUP FOR RE-ID MODEL EVALUATION WITH PSEUDO-OCCLUSIONS

We mask the GT tracks in the nuScenes validation split to create pseudo-occlusions to evaluate our Re-ID model independently. In every evaluated sample, all future tracks are masked for random durations to create diverse pseudo-occlusions. The durations range from 1.5s to each track's maximal length (12.5s at most) such that each future tracklet has at least one last pose visible. The history tracklet in each sample is also randomly deprecated such that it has at least one pose left as input and is at most 2.5 seconds long. The distribution of pseudo-occlusions is shown in Fig. 9. The number of matching candidate tracklets ranges from 2 to 65, as shown in Fig. 8.

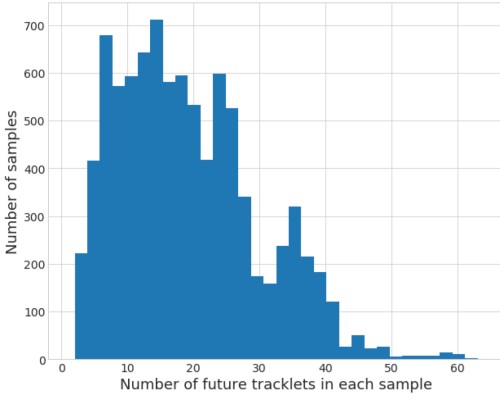
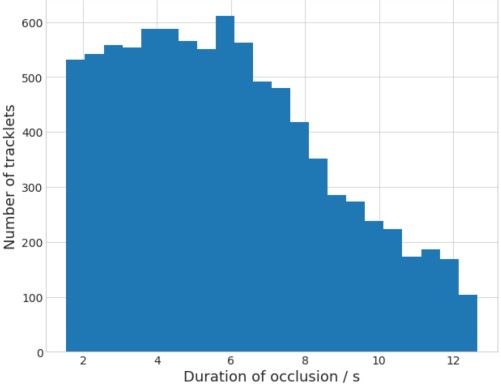

Figure 8: The distribution of the number of candidate tracklets in Re-ID model evaluation with pseudo-occlusions. The candidate tracklet number ranges from 2 to 65.

Figure 9: The distribution of the pseudo occlusion durations in Re-ID model evaluation. The occlusion duration ranges from 1.5 seconds to 12.5 seconds.

