# OpenReview forum: "Offline Tracking with Object Permanence"
_ICLR.cc/2024/Conference — Submitted to ICLR 2024_

### Official Review · Reviewer_SgKx · 2023-10-28

**Soundness:** 2 fair
**Presentation:** 2 fair
**Contribution:** 1 poor
**Rating:** 3
**Confidence:** 4

**Summary:**

The paper introduces an offline tracking method utilizing point clouds, specifically designed to handle challenging heavy occlusions on vehicles. This method comprises three key components: a tracker, a Re-ID module for linking trackless segments before and after occlusions, and a track completion module that interpolates missing tracks caused by occlusions. The study showcases the method's efficacy in tracking objects even under conditions of occlusion.

**Strengths:**

The paper proposes a method to handle the occluded object tracking.

**Weaknesses:**

The paper has limited novelty. The tracker, Re-ID, and track completion components all employ established techniques, resulting in a relatively straightforward solution.

**Questions:**

1. In tracklet association, there appears to be a division between the utilization of map and motion data. However, it's worth considering their complementary contributions. For instance, map information could enhance the accuracy of motion association. Separating these aspects might lead to a loss of crucial information.
2. Appearance features could serve as a pivotal factor in tracklet association. Even when an object becomes temporarily occluded and untrackable, its re-emergence could still benefit from utilizing appearance data for accurate association.
3. Considering the use of map information, it's important to assess its impact on association when an object undergoes occlusion and a simultaneous lane change. This scenario introduces an additional layer of complexity that warrants thorough investigation.

---

> ### Author Response · Authors · 2023-11-23
> **Addressing the questions**
>
> Thanks a lot for your questions! Here are some concise responses to them.
>
> 1. Indeed, the fusion of the two features can theoretically improve performance. That is also exactly what we did. In the map branch of the Re-ID model, the map feature is fused with the motion feature via spatial attention [1]. So the map branch takes both motion information and map information as inputs.
>
> 2. Thanks for your suggestion! We have also tried to extract appearance features from point clouds for Re-ID. We used the same technique for point cloud aggregation as in [2], only that we used CenterPoint [3] as the feature extractor. Then we applied contrastive learning to learn the appearance affinity between the appearance features of tracklets in a latent space. However, the result is not satisfying so we eventually abandoned this. We think this is because the model can only implicitly extract the object size information from the point cloud and learns the affinity based on this, which is not accurate. The location and motion information in point clouds are also provided by the motion feature. That is also the reason why the appearance feature extracted from the point cloud is mainly used to refine the bounding box size in [2,4]. However, we are planning to explore a more effective way to extract it so that it can help tracklet association. A more trivial way is to use extract appearance feature from camera images, which contains much richer appearance information for Re-ID.
>
> 3. Indeed, lane change under occlusion would be an interesting scenario for the evaluation of our model. We will add this to our evaluation in the future!
>
> [1] Ming Liang, Bin Yang, Rui Hu, Yun Chen, Renjie Liao, Song Feng, and Raquel Urtasun. Learning lane graph representations for motion forecasting. In ECCV, pp. 541–556. Springer, 2020.
>
> [2] Bin Yang, Min Bai, Ming Liang, Wenyuan Zeng, and Raquel Urtasun. Auto4d: Learning to label 4d objects from sequential point clouds. arXiv preprint arXiv:2101.06586, 2021.
>
> [3] Tianwei Yin, Xingyi Zhou, and Philipp Krahenbuhl. Center-based 3d object detection and tracking. In CVPR, pp. 11784–11793, 2021.
>
> [4] Charles R Qi, Yin Zhou, Mahyar Najibi, Pei Sun, Khoa Vo, Boyang Deng, and Dragomir Anguelov. Offboard 3d object detection from point cloud sequences. In CVPR, pp. 6134–6144, 2021.

---

### Official Review · Reviewer_tT1W · 2023-10-31

**Soundness:** 3 good
**Presentation:** 3 good
**Contribution:** 2 fair
**Rating:** 3
**Confidence:** 3

**Summary:**

This paper proposes a non-causal MOT technique for labelling large datasets in autonomous driving without human intervention. The technique reconsiders three well-known steps: 1) finding tracklets using causal MOT, 2) associating those tracklets by Re-ID and 3) trajectory completion compensating occlusions. The authors propose novel neural models for each step, including bipartite matching for association. The paper is well structured and readable; the literature research at least shows recent work to compare with.

**Strengths:**

The paper shows an elaborated approach to noncausal MOT. The neural models used seem innovative and novel and the attempt to combine tracklet association with a priori knowledge of lane maps in one end-to-end framework is promising.

**Weaknesses:**

Unfortunately, the paper is difficult to understand for the reader. Many details are presented in a course to a more detailed manner.
After reading the paper, it is not clear what exactly the contribution is. The idea of using object permanence for tracking has been introduced previously (Tokmakov et al.). The three steps of MOT are not new (Zhang, Li, Nevatia, Global Data Association for Multi-Object Tracking Using Network Flows, 2012).
The neural models seem novel. However, the claim in the abstract that the new models improve IDS by 45% is not confirmed by the experimental results. Table 2 shows the causal MOT of Wang et al., 2021 with 109 IDS better than the proposed method with 147+ depending on the version. The performance measures do not show significant evidence for improvement over the causal MOT approach, neither for table 1 nor for table 3.

**Questions:**

Considering the result of your experiments, as shown in table 1 - 3, why is the proposed method superior to the compared causal trackers?
Why is the proposed method better suited for offline labelling than the online trackers?
Does the proposed methods with approx. AMOTP 0.603 and IDS 145 allow the labelling of datasets?

---

> ### Comment · Reviewer_tT1W · 2023-12-05
>
> My advise: Select carefully the experiments (as in Table 3) to show significant evidence of your approach if tracking performance is not the main aim but occlusion. I invite the authors to improve the framework following the advice of the reviewers and consider resubmission.

---

### Official Review · Reviewer_GvzN · 2023-11-01

**Soundness:** 2 fair
**Presentation:** 3 good
**Contribution:** 2 fair
**Rating:** 5
**Confidence:** 4

**Summary:**

To track occluded objects, this paper proposes an offline tracking framework, including an online tracker to generate initial tracklets, a reid module to associate tracklets, and a track completion module to complete the fragmented tracks. Through aggregating and decoding outputs from several different encoders, the track completion model will output the final refined trajectory. Experiments are performed on nuScenes dataset with different evaluation setups.

**Strengths:**

1. The proposed framework is novel, which embeds both the motion and lane map to obtain the final matching matrix, and also fuses the time query embeddings to implement the trajectory regression.
2. This paper evaluates the proposed method under different evaluation setups, which demonstrate the effectiveness of the method more clearly in addressing occlusion situations.

**Weaknesses:**

1. Experiments are only performed on the validation split of nuScene dataset, and compared with few SOTA methods, which lack evidence of its effectiveness to some extent.

2. Based on Table1&2, the effect of the motion embedding doesn't seem obvious. Besides, there are no more ablation studies to show the effectiveness of other designs, like the impact of training with augmented GT data instead of tracker outputs, the impact of embedding time query, and so on.

3. The three modules in the proposed framework are separate from each other, rather than a unified end-to-end model, which makes the proposed framework a bit complicated.

**Questions:**

The issues I am concerned about are listed in order in the above "Weaknesses". I'll change my rating if the authors explain the first two issues well.

---

### Official Review · Reviewer_S5GM · 2023-11-01

**Soundness:** 2 fair
**Presentation:** 2 fair
**Contribution:** 1 poor
**Rating:** 5
**Confidence:** 3

**Summary:**

This paper proposes an offline tracking framework with object permanence to reduce the expensive labor cost in labeling large-scale autonomous driving datasets. The proposed approach can be briefly summarised as several steps: 1) applying the off-the-shell detector and tracker to generate initial tracklets; 2) using the Re-ID module for tracklet association; 3) employing the track completion module for trajectory completion. Specifically, the effectiveness of the model is validated on the nuScenes validation split.

**Strengths:**

- This paper aims to solve an essential problem in autonomous driving dataset labelling.
- Quantitative and qualitative results show some superiority of the proposed approach over the compared approaches.

**Weaknesses:**

- The technical contribution is limited. The proposed approach heavily relies on off-the-shell detectors/tracker, and are inspired from existing approaches a lot (especially for the track completion module), which seems not significant enough as the main contributions by considering the object permanence conception had already been proposed in previous works [1].
- Missing details about the training hyper-parameters for reproduction.
- The proposed approach does not show significant improvements over the compared Immortal tracker (See Tables 1-3). Is there any specifically design in Immortal tracker making the comparison unfair? Otherwise, it cannot effectively show the superiority of the proposed approach in this paper.

[1] Pavel Tokmakov, Jie Li, Wolfram Burgard, and Adrien Gaidon. Learning to track with object permanence. In ICCV, pp. 10860–10869, 2021.

**Questions:**

- The technical contribution is limited. The proposed approach heavily relies on off-the-shell detectors/tracker, and are inspired from existing approaches a lot (especially for the track completion module), which seems not significant enough as the main contributions by considering the  object permanence conception had already been proposed in previous works [1].
- Missing details about the training hyper-parameters for reproduction.
- The proposed approach does not show significant improvements over the compared Immortal tracker (See Tables 1-3). Is there any specifically design in Immortal tracker making the comparison unfair? Otherwise, it cannot effectively show the superiority of the proposed approach in this paper.

[1] Pavel Tokmakov, Jie Li, Wolfram Burgard, and Adrien Gaidon. Learning to track with object permanence. In ICCV, pp. 10860–10869, 2021.

---

> ### Comment · Reviewer_S5GM · 2023-12-04
>
> Thanks for the detailed feedback. I agree with the other reviewers that the technical contributions in this paper are not significant enough. The newly added results in Table 2 are interesting, showing larger improvements over the baselines compared to the results posted in the main paper. It is suggested to more clearly illustrate or highlight the contributions (maybe using some figures for pipeline comparison between the proposed framework and previous ones) and include all newly added results (especially Table 2) in the main paper for effectiveness demonstration. Currently, I am still keeping the original rating since the paper is not good enough for acceptance.

---

### Author Response · Authors · 2023-11-23
**Addressing a common and important concern -- 1 : Weak/unconvincing results on validation split over causal tracker**

Dear Reviewers,


Thanks a lot for all the questions and suggestions!


**I summarized two important common concerns from reviewers. I will address them globally as they are proposed by multiple reviewers.**

### Concern 1: Weak/unconvincing results on validation split over causal tracker.
### Response:
1. Indeed, our previous results did not show much superiority over the causal Immortal Tracker. We have updated the Re-ID result which is now benchmarked on the nuScenes test split. The result is also publicly available on the official website. The updated result shows more superiority. Here, we compare our offline Re-ID result on the vehicle classes with the top-ranking trackers that also use CenterPoint detections [1].


| Method | Overall | Car | Bus | Truck | Trailer |
| :---: | :---: | :---: | :---: | :---: | :---: |
| CenterPoint [1] | 69.8 | 82.9 | 71.1 | 59.9 | 65.1 |
| SimpleTrack [2] | 70.0 | 82.3 | 71.5 | 58.7 | 67.3 |
| UVTR [3] | 70.1 | 83.3 | 67.2 | 58.4 | $\mathbf{7 1 . 6}$ |
| Immortal Tracker [4] | 70.5 | 83.3 | 71.6 | 59.6 | 67.5 |
| NEBP [5] | 70.8 | 83.5 | 70.8 | 59.8 | 69 |
| 3DMOTFormer++ [6] | 72.3 | 82.1 | 74.9 | 62.6 | 69.6 |
| ShaSTA [7] | 73.1 | 83.8 | 73.3 | $\mathbf{6 5}$ | 70.4 |
| $\textbf{Offline Re-ID}$ (Motion + Map) | $\mathbf{7 3 . 4}$ | $\mathbf{8 4 . 2}$ | $\mathbf{7 5 . 1}$ | 64.1 | 70.3 |

Table 1: Comparison of **AMOTA $\uparrow$** scores over the SOTA methods using CenterPoint [1] detections on the nuScenes test split (official nuScenes setup).



| Method | AMOTP $\downarrow / \mathbf{m}$ | Recall $\uparrow$ | MOTA $\uparrow$ | IDS $\downarrow$ |
| :---: | :---: | :---: | :---: | :---: |
| CenterPoint [1] | 0.596 | 73.5 | 59.4 | 340 |
| SimpleTrack [2] | 0.582 | 73.7 | 58.6 | 259 |
| UVTR [3] | 0.636 | $\mathbf{7 4 . 6}$ | 59.3 | 381 |
| Immortal Tracker [4] | 0.609 | 74.5 | 59.9 | 155 |
| NEBP [5] | 0.598 | 74.1 | $\mathbf{6 1 . 9}$ | $\mathbf{9 3}$ |
| 3DMOTFormer++ [6] | 0.542 | 73.0 | 58.6 | 210 |
| ShaSTA [7] | 0.559 | 74.3 | 61.2 | 185 |
| $\textbf{Offline Re-ID}$ (Motion + Map) | $\mathbf{0 . 5 3 2}$ | 74.2 | 61.3 | 204 |

Table 2: Comparison of MOT metrics over the SOTA methods using CenterPoint [1] detections on the nuScenes test split (official nuScenes setup).


2. More importantly, our main aim is to recover occlusions (which improves tracking results to some extent), rather than only improve tracking performance. The improvement brought by our method over the online tracking results is upper bound by the number and length of occlusions in the dataset. The occlusion scenarios are common but still a relatively small portion of the dataset which limits the performance of our model.

    However, since the existing offline labeling methods follow the 'Detection-Tracking-Refinement' pipeline [9-12], our model can be easily used as a useful plugin for offline auto labeling to improve tracking by recovering occlusions. Theoretically, it can improve any trackers that suffer from occlusions. We demonstrate this by applying our offline model to multiple online trackers and show the improvements.


| Metrics | AMOTA $\uparrow$ | AMOTA $\uparrow$ | AMOTP $\downarrow / \mathbf{m}$ | AMOTP $\downarrow / \mathbf{m}$ | IDS $\downarrow$ | IDS $\downarrow$ | Recall $\uparrow$ | Recall $\uparrow$ |
| :---: | :---: | :---: | :---: | :---: | :---: | :---: | :---: | :---: |
| Occlusion Recovery | w/o | w | w/o | w | w/o | w | w/o | w |
| CenterPoint [1] | 70.2 | $\mathbf{7 2 . 4}$ | 0.634 | $\mathbf{0 . 6 1 5}$ | 254 | $\mathbf{1 8 3}$ | 73.7 | $\mathbf{7 4 . 5}$ |
| SimpleTrack [2] | 70.0 | $\mathbf{7 1 . 0}$ | 0.668 | $\mathbf{0 . 6 2 9}$ | 210 | $\mathbf{1 7 0}$ | 72.5 | $\mathbf{7 2 . 9}$ |
| VoxelNet [8] | 69.6 | $\mathbf{7 0 . 6}$ | 0.710 | $\mathbf{0 . 6 6 5}$ | 308 | $\mathbf{2 3 0}$ | 72.8 | $\mathbf{7 2 . 9}$ |
| ShaSTA [7] | 72.0 | $\mathbf{7 2 . 6}$ | 0.612 | $\mathbf{0 . 5 9 3}$ | 203 | $\mathbf{1 7 4}$ | 73.0 | $\mathbf{7 5 . 3}$ |

Table 3: Joint evaluation on the nuScenes validation split (All-boxes setup). $\textbf{w}$: with offline Re-ID and track completion. $\textbf{w/o}$: original results without any refinement.

[1] Yin et al. Center-based 3d object detection and tracking.
In CVPR,  2021.

[2] Pang et al. Simpletrack: Understanding and rethinking 3d multi-object tracking. In ECCV, 2022.

[3] Li et al. Unifying voxel-based representation with transformer for 3d object detection. Advances in Neural Information Processing Systems,  2022.

[4] Wang et al. Immortal tracker:
Tracklet never dies. arXiv preprint arXiv:2111.13672, 2021.

[5] Victor Garcia Satorras and Max Welling. Neural enhanced belief propagation on factor graphs. In
International Conference on Artificial Intelligence and Statistics. PMLR, 2021.

[6] Ding et al. 3dmotformer:
Graph transformer for online 3d multi-object tracking. In ICCV, 2023.

[7] Sadjadpour et al. Shasta: Modeling shape and spatio-temporal affinities for 3d multi-object tracking. RA-L, 2023

---

> ### Author Response · Authors · 2023-11-23
> **Adding a missing citation**
>
> [8] Chen et al. Voxelnext: Fully sparse voxelnet for 3d object detection and tracking. In CVPR, 2023.

---

### Author Response · Authors · 2023-11-23
**Addressing a common and important concern -- 2: Limited contribution or novelty and heavily relying on existing methods.**

Dear Reviewers,

Thanks a lot for all the questions and suggestions!

**I summarized two important common concerns from reviewers. I will address them globally as they are proposed by multiple reviewers.**

### Concern 2: Limited contribution or novelty/ Heavily relying on existing methods.
### Response:
I will divide this concern into two levels and address them separately.
1. From reviewers [**S5GM**](https://openreview.net/forum?id=8tWOUmBHRv&noteId=o2DYqsJ6lZ) and [**tT1W**](https://openreview.net/forum?id=8tWOUmBHRv&noteId=NVlxpeFUPL), the concept of Object Permanence has already been applied for MOT in [1].

    Indeed, the concept itself is not novel in MOT. However, we claim that object permanence is still underexplored previously. In [1], even though the tracker can hallucinate the occluded motion to some extent, it fails to capture the non-linear motion behind occlusions. The authors in [1] only assumed the occluded objects keep moving with constant velocity under occlusion. That is why they only use pseudo-labels with constant velocity to train the model. Even if the authors tried to use GT labels to supervise the model for occlusion cases, it yielded worse performance than using the constant-velocity pseudo-labels. In summary, the model in [1] is indeed aware of object permanence, but it still simply relies on the motion heuristic of constant velocity, which makes it difficult for the model to accurately infer the occluded motion.

    Our model, on the other hand, not only is aware of the occlusion but also can infer the non-linear motion behind the occlusion. Because we used the lane map as a prior to guide the vehicle motion under occlusions. The effectiveness is proved by showing the superiority of our model over the Kalman filter-based models with a constant velocity transition model. This is also one of our novelties, we are currently the only one using the lane map for MOT.

2. From reviewers [**S5GM**](https://openreview.net/forum?id=8tWOUmBHRv&noteId=o2DYqsJ6lZ),  [**tT1W**](https://openreview.net/forum?id=8tWOUmBHRv&noteId=NVlxpeFUPL), and [**SgKx**](https://openreview.net/forum?id=8tWOUmBHRv&noteId=l4jONW0I24), the three steps are not new, and each submodule is inspired or built upon off-the-shelf or established methods.

    The reason why we design our model this way is because we want our model to be as compatible as possible with the existing offline autolabeling methods. The existing offline labeling methods follow the 'Detection-Tracking-Refinement' pipeline [2-5], and all use off-the-shelf trackers (or modified from them). Therefore, our model can be simply integrated into them as a plugin without any further adjustments. Furthermore, our model can take the output results from any off-the-shelf trackers, meaning that people can simply change the off-the-shelf tracker for customization or better performance.

    Lastly, our model did use some standard building blocks from other prediction methods, such as the spatial attention in [6].

[1] Pavel Tokmakov, Jie Li, Wolfram Burgard, and Adrien Gaidon. Learning to track with object permanence. In ICCV, pp. 10860–10869, 2021.

[2] Bin Yang, Min Bai, Ming Liang, Wenyuan Zeng, and Raquel Urtasun. Auto4d: Learning to label
4d objects from sequential point clouds. arXiv preprint arXiv:2101.06586, 2021.

[3] Lue Fan, Yuxue Yang, Yiming Mao, Feng Wang, Yuntao Chen, Naiyan Wang, and Zhaoxiang Zhang.
Once detected, never lost: Surpassing human performance in offline lidar based 3d object detection. ICCV, 2023.

[4] Charles R Qi, Yin Zhou, Mahyar Najibi, Pei Sun, Khoa Vo, Boyang Deng, and Dragomir Anguelov.
Offboard 3d object detection from point cloud sequences. In CVPR, pp. 6134–6144, 2021.

[5] Tao Ma, Xuemeng Yang, Hongbin Zhou, Xin Li, Botian Shi, Junjie Liu, Yuchen Yang, Zhizheng
Liu, Liang He, Yu Qiao, et al. Detzero: Rethinking offboard 3d object detection with long-term
sequential point clouds. ICCV, 2023.

[6] Ming Liang, Bin Yang, Rui Hu, Yun Chen, Renjie Liao, Song Feng, and Raquel Urtasun. Learning
lane graph representations for motion forecasting. In ECCV, pp. 541–556. Springer, 2020.

---

### Meta-Review · Area_Chair_cJv7 · 2023-12-06

**Metareview:**

The paper was reviewed by 4 experts, with ratings 3355. The major concerns of the reviewers were:

1. limited technical contribution - reliance on off-the-shelf components. [S5GM, tT1W, SgKx]
2. missing details about hyperparmeters [S5GM]
3. no significant improvement compared to other trackers - what is the advantage of the proposed approach? [S5GM, tT1W]
4. experiments only on the validation split of nuScene [GvzN]
5. effect of motion embedding is not obvious. no ablation studies on other designs [GvzN]
6. pipeline uses 3 separate modules, which makes it not end-to-end. [GvzN]
7. IDF claim is unfounded [tT1W]
8. why is there a division between the utilization of map and motion data. Could they provide complementary information? [SgKx]
9. could appearance features be helpful? [SgKx]
10. what is the impact of map information when the object undergoes occlusion and lane change simultaneously? [SgKx]

The authors wrote a response, but the response did not assuage the concerns of the reviewers, in particular the limited technical contribution and the weak experiment results compared to SOTA. The AC agrees with these concerns and thus recommends reject.

**Justification For Why Not Higher Score:**

technical contribution is weak. experiment results are mixed.

**Justification For Why Not Lower Score:**

n/a

---

### Decision · Program_Chairs · 2024-01-16

Reject